# Molecular Insights into IQSEC2 Disease

**DOI:** 10.3390/ijms24054984

**Published:** 2023-03-05

**Authors:** Nina S. Levy, Veronika Borisov, Orit Lache, Andrew P. Levy

**Affiliations:** Technion Faculty of Medicine, Technion Israel Institute of Technology, Haifa 3200003, Israel

**Keywords:** IQSEC2, Arf6-GTP, heat shock

## Abstract

Recent insights into IQSEC2 disease are summarized in this review as follows: (1) Exome sequencing of IQSEC2 patient DNA has led to the identification of numerous missense mutations that delineate at least six and possibly seven essential functional domains present in the IQSEC2 gene. (2) Experiments using IQSEC2 transgenic and knockout (KO) mouse models have recapitulated the presence of autistic-like behavior and epileptic seizures in affected animals; however, seizure severity and etiology appear to vary considerably between models. (3) Studies in IQSEC2 KO mice reveal that IQSEC2 is involved in inhibitory as well as stimulatory neurotransmission. The overall picture appears to be that mutated or absent IQSEC2 arrests neuronal development, resulting in immature neuronal networks. Subsequent maturation is aberrant, leading to increased inhibition and reduced neuronal transmission. (4) The levels of Arf6-GTP remain constitutively high in IQSEC2 knockout mice despite the absence of IQSEC2 protein, indicating impaired regulation of the Arf6 guanine nucleotide exchange cycle. (5) A new therapy that has been shown to reduce the seizure burden for the IQSEC2 A350V mutation is heat treatment. Induction of the heat shock response may be responsible for this therapeutic effect.

## 1. Introduction

IQSEC2 is an X-linked gene that is associated with intellectual disability, autism, and epilepsy [1]. Mutations in IQSEC2 account for approximately 2% of patients with ID and epilepsy referred for exome sequencing [2]. Treatment for these patients is lacking and seizure control is difficult to attain. With the advent of new animal and cellular models for studying the disease, new insights have been gained into the normal function of IQSEC2 and the pathways that may be aberrant in its absence or when its function is altered. In addition, a number of therapeutic strategies have been described. This review will summarize new developments in IQSEC2 research and treatment and discuss promising future directions.

## 2. IQSEC2 Function

IQSEC2 is a member of a subfamily of homologous proteins (IQSEC1, IQSEC2, and IQSEC3) known as guanine nucleotide exchange factors, or GEFs. These GEFs exchange GTP for GDP on another family of proteins known as small GTPases or small G proteins (see Um et al. [3], Petersen et al. [4], and D’Souza and Casanova [5] for review). The IQSEC proteins are GEFs for the six-member Arf family of small G proteins Arf1-6. The Arfs were first identified as a cellular activity required for ADP-ribosylation of Gαs by cholera toxin, a process by which it exerts its toxic effect. Much of the current research on Arfs is not related to ADP-ribosylation. Rather, these small G proteins are best known for participating in membrane trafficking, lipid transformation, and reorganization of the actin cytoskeleton. IQSEC2 is thought to interact most specifically with Arf6 [6], which is the only Arf that regulates the recycling of endosomes and receptors to and from the plasma membrane.

The role of IQSEC2 at excitatory synapses has been the focus of numerous studies. The combined work of a number of laboratories supports the following mechanism of action for IQSEC2 [7,8,9,10]: Glutamate release from presynaptic neurons leads to NMDA receptor activation and the influx of calcium ions in postsynaptic neurons. NMDA receptors are also complexed to PSD95 and are therefore in close proximity to IQSEC2. Calcium binds to calmodulin present on the IQ domain of IQSEC2, leading to its dissociation from IQSEC2 and to activation of the GEF catalytic Sec7 domain of IQSEC2. Arf6-GDP undergoes the exchange of GDP for GTP by the Sec7 domain of IQSEC2, thereby leading to the activation of Arf6. Arf6-GTP mediates the activation of downstream effectors such as phospholipase D, phosphatidylinositol-4-phosphate 5-kinases, and Ras-related C3 botulinum toxin substrate 1 (Rac1), leading to changes in membrane trafficking and actin dynamics. Arf6-GTP is inactivated by a GTPase-activating protein (GAP). Myers et al. [9] showed that both active IQSEC2 and JNK activity are required for the downstream removal of AMPA receptors in hippocampal neurons following excitatory stimulation. Ultimately, IQSEC2 is responsible for promoting the growth and development of dendritic spines, axonal elongation, and branching in postsynaptic neurons, all necessary for the proper development of cognition and learning [11].

## 3. IQSEC2 Structure

IQSEC2 is a 1488 amino acid protein containing 6 known functional domains (see Figure 1). These include: (1) an N terminal coiled-coil (CC) domain that mediates protein self-association; (2) an IQ-like domain that binds apo-calmodulin and allosterically influences the Sec7 domain; (3) a Sec 7 domain responsible for GDP–GTP exchange on Arf6; (4) a pleckstrin homology (PH) domain which is thought to be involved in IP3 signaling and localization to the plasma membrane; (5) a proline-rich (PR) domain which is known to bind to the insulin receptor tyrosine kinase substrate of 53 kD (IRsp53), recently shown to control plasma membrane shape [12] and may be involved in dendrite formation; and (6) a PDZ domain which mediates binding to PSD95, an important member of the postsynaptic density responsible for anchoring proteins to the cytoskeleton and mediating signal transduction following excitatory stimulation.

Since the discovery of IQSEC2 disease [1], over 120 new mutations have been reported in IQSEC2 patients (see Appendix A) [13,14,15,16,17]. The majority of cases are null mutations, in which the gene has been deleted or a nonsense codon has been created due to a point mutation or a frameshift mutation. Nonsense mutations generally result in mRNA decay or a truncated protein that is rapidly degraded. However, 37 independent missense mutations have been reported which are concentrated in multiple functional domains present in IQSEC2 (see Figure 1 and Appendix A).

The majority of missense mutations fall within four functional domains (CC, IQ, Sec7, and PH). Two additional binding domains are certain to be critical (PR and PDZ) as well, but no missense mutations have been reported in these areas as they are quite small (4 and 10 aa respectively). However, their importance may be seen in the case of a missense mutation, R1402T [18], that falls quite close to the PR domain and may affect the binding of IRsp53. One other alteration at aa 1468 and two at aa 1474 create frameshift mutations that terminate after 27, 21, and 133 aa respectively, resulting in the specific abrogation of the PDZ domain. In addition, new documented mutations in IQSEC2 reveal a cluster of four single nucleotide changes just C-terminal to the PH domain, in an area that has not been categorized to date. This may represent a new functional domain that is yet to be characterized. Interestingly, IQSEC1 and IQSEC3 have a second CC domain in this region [3]. Although the homology for the CC domain is not conserved in IQSEC2, this area may contain some common secondary structure of importance in all three family members. Three additional mutations, S257N, R563Q, and D706S, do not fall in any defined domain. These mutations may disrupt intra-protein folding involved in the allosteric effect of the IQ domain on the Sec7 domain. In vitro expression of these mutations might shed light on this possibility.

**Figure 1 ijms-24-04984-f001:**
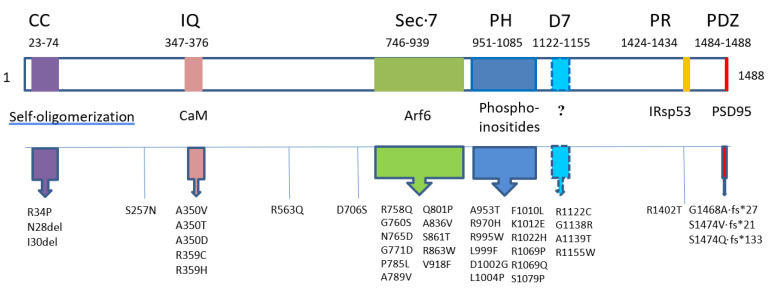
Mutations affecting specific functional domains in IQSEC2. The 1488 aa isoform of the IQSEC2 protein is shown as an elongated rectangle. Functional domains (colored regions) are listed on top of the diagram with amino acid boundaries indicated underneath. They are coiled-coil (CC), IQ-like (IQ), Sec7, pleckstrin homology (PH), proline-rich (PR), PDZ binding motif (PDZ), and a potentially new 7th domain (D7). Domain-binding entities are listed below the diagram. Missense mutations, in-frame deletions (del), and frameshift mutations creating a termination codon (fs*) are listed underneath the relevant domains. R1402 [18] is in close proximity to and likely affects the PR domain. R1122C [15], G1138 [16], A1139 [14], and R1155W [14] define domain D7. S257N [19], R563Q [17], and D706S [16] do not fall in any previously described domain and their mode of pathogenicity is not known. One possibility is that they disrupt the allosteric influence of the IQ domain on Sec7 activity.

In summary, IQSEC2 appears to contain six or possibly seven functional domains which are absolutely required for normal development and are likely to be involved in multiple protein–protein or protein–membrane interactions. Frank et al. [20] have shown that PSD95 (along with IQSEC2) is found in 1.5 MDa complexes in the mouse forebrain. Most of the work on IQSEC2 has been focused on its physical association with the NMDA receptor. However, the vast majority (97%) of IQSEC2 is not found in complexes with NMDA receptors, indicating that we have only scratched the surface regarding the scope of IQSEC2 interactions [20].

## 4. Recapitulation of IQSEC2 Disease in Mice

Four mouse models of IQSEC2 disease have been studied. Three are independent knockout (KO) models and one is a transgenic A350V mutation model. The mice used by Mehta et al. [21] were created by introducing a 17 bp deletion into exon 3 of the IQSEC2 gene and the KO mice were maintained on a C57BL6/JJcl and 129+Ter/ScJcl hybrid background. The model used by Sah et al. [22] introduced a single nucleotide deletion and translational frameshift in IQSEC2 exon 7 and the mice were maintained on a mixed C3HeB/FeJ and C57BL/6NJ background. Jackson et al. [23] generated a KO strain which deleted exon 3 and the mice were maintained on a C57BL/6N-Hsd background. All of the KO models resulted in the generation of a premature stop codon with no detectable IQSEC2 protein expression. The A350V transgenic mouse strain [10] was created using CRISPR/Cas9 targeting of the murine wild-type IQSEC2 gene generating an A350V mutation identical to that found in the human index case in which the codon GCT (Ala) at amino acid 350 is mutated to GTT (Val) with an additional AGG to CGT silent mutation (R349). The mutant mice were maintained on a C57Bl/6J background. Mutant A350V protein was found to be expressed in the brain by Western blot analysis (see Appendix A).

All of the models exhibited decreased fertility, hyperactivity, defects in social interactions, and abnormalities in the electrophysiology of isolated neurons (see discussion below). Anxiety was found in two of the KO models [22,23]. Cognitive function was looked at in two models [10,23] and found to be somewhat decreased although this parameter was not studied in depth. Three of the models looked at seizure activity [10,22,23]. The timing and severity of the seizures varied considerably between models. Sah et al. [22] reported that the susceptibility of the KO mice to induced seizures was suppressed and the KO mice were observed to have seizures only on long-term video. However, a post-mortem examination showed evidence of lethal seizures and mortality was 100% in males by approximately 6 months of age. Jackson et al. [23] reported that KO males and heterozygous females had spontaneous seizures (52% and 46%, respectively) beginning on days 29 and 23 after birth, respectively. Mortality was 20% in males and 31% in females. In the A350V model, male mice suffered spontaneous lethal seizures between days 16–20 after birth [24]. Mortality was 43% for males and 20% for females. The variability seen in seizure activity among all of these studies may be due to differences in seizure thresholds of the different genetic strains used or to differences in mutations (KO versus missense). Although IQSEC2 does not escape X chromosome inactivation in mice, the mechanisms governing inactivation can differ between strains, making it difficult to compare results. It may be that finding appropriate drugs to treat IQSEC2-related epilepsy will require personalized systems such as iPSC cells.

With regard to therapeutic treatments, Mehta et al. [21] were able to reverse electrophysiological and behavioral deficits by infecting medial prefrontal cortex (mPFC) neurons in IQSEC2 KO mice with an adeno-associated virus (AAV) vector encoding IQSEC2 under the control of the EF1alpha short (EFS) promoter. The social discrimination deficits seen in the A350V mice were rescued with a single dose of PF-4778574, a positive AMPAR modulator [25]. Seizure activity, electrophysiology, and social deficits in the A350V mice were rescued with heat treatment, as will be discussed below [26].

## 5. IQSEC2 Dosage

Looking at the clinical data in general, it can be seen that the disease is more severe in males than females [17]. This is logical considering that IQSEC2 is known to escape X inactivation in humans, and therefore female carriers can benefit from one normal IQSEC2 allele. Paradoxically, the levels of IQSEC2 are regulated in females such that overall the levels of IQSEC2 are approximately the same in both sexes [27]. Another report showed that IQSEC2 is expressed at higher levels in males than in females in the brain cortex [28]. The reason for this discrepancy is not clear; however, it seems clear that the level of IQSEC2 is tightly regulated, at least in females. Differences in this regulatory mechanism among individuals may explain why related females heterozygous for the same mutation have differing severity of the disease [17].

Evidence that reduced levels of IQSEC2 are detrimental comes from a study by Madrigal et al. [29], who reported on a family with a splice site mutation in IQSEC2. The percent of aberrant splicing among family members correlated with the severity of the disease, supporting the tenet that disease status is determined by the degree to which IQSEC2 levels are diminished. In vitro studies using inhibitory RNA have shown that a reduction in IQSEC2 leads to abnormalities such as disturbed growth and morphology of developing neurons in cell culture [11] and constitutively activated ARF6-GTP in neuronal cultures [8]. However, the pathogenicity of increased levels of IQSEC2 is less clear. Microduplications in patients with intellectual disability have been found which encompass three disease genes, TSPYL2, KDM5C, and IQSEC2 [27]. These patients’ symptoms may be due to increased IQSEC2, although the contribution of increased IQSEC2 alone is difficult to determine from this study. Myers et al. [9] overexpressed mCherry-tagged IQSEC2 in rat hippocampal slices and showed that the fusion proteins localized to excitatory synapses, similar to endogenous IQSEC2. Brown et al. [7] showed that overexpressing wild-type IQSEC2 in neurons leads to increased neural transmission. Hinze et al. [11] reported that overexpressing wild-type IQSEC2 in IQSEC2 KO neurons altered dendrite and spine morphology compared to wild-type cells. The last two studies suggest that overexpression of IQSEC2 could be harmful; however, there was no quantitation of the levels of IQSEC2 induced in these experiments, making it difficult to know the characteristics of the dose–response curve. Interestingly, as noted above, Mehta et al. [21] used gene therapy to reverse social and electrophysiological deficits in IQSEC2 KO mice. Although quantitation of IQSEC2 expression in treated KO animals was not reported, the authors did investigate the effect of gene therapy in WT animals. These studies showed that overexpression of IQSEC2 in the mPFC of wild-type animals did not affect social behaviors. In summary, it may be that a threshold level of IQSEC2 is required for normal development, below which defects may occur. However, moderately increased levels may be well tolerated. Transgenic mice containing several copies of the IQSEC2 gene under inducible control might shed light on this question, which has important implications for the potential use of AAV vectors for human gene therapy.

## 6. Role of IQSEC2 in Inhibitory Neural Transmission

Although expressed throughout the brain, the highest levels of IQSEC2 have been found in the hippocampus, with expression previously thought to be restricted to the postsynaptic density of excitatory neurons [6]. Prior reports of IQSEC2 electrophysiological function measured excitatory currents in wild-type hippocampal neurons transfected with different IQSEC2 variants, resulting in the overall mechanism of action described above. IQSEC3, another neuron-specific member of the IQSEC family, was reported to be associated with gephyrin [3], a molecule exclusively present in inhibitory neurons. Transfection experiments designed to increase or decrease the expression of IQSEC3 resulted in greater or lesser inhibitory transmission, respectively. This led to the presumption that IQSEC2 is mainly involved in mediating excitatory signaling while IQSEC3 is involved in mediating inhibitory transmission.

Reports using the IQSEC2 KO mouse models paint a new picture. Mehta et al. [21] have shown that both excitatory as well as inhibitory synaptic transmissions are impaired in their knockout (KO) model of IQSEC2. Patch clamp recordings were performed on brain slices containing pyramidal neurons in layer 5 of the mPFC in P14–P19 day-old mice. The researchers found that the frequency, but not amplitude, of miniature excitatory (mEPSCs) and inhibitory (mIPSCs) postsynaptic currents were significantly decreased in IQSEC2 KO mice. In addition, it was found that both excitatory (AMPA and NMDA) and inhibitory (GABA) currents were decreased in response to evoked changes in the membrane potential of neurons from IQSEC2 KO mice.

The study by Sah et al. [22] also reveals a role for IQSEC2 as an important modulator of inhibitory neurotransmission. This group measured synaptic transmission in dissociated KO hippocampal neurons from P1–P2 day-old mice after 12 days in culture. The authors transfected the cultured neurons with an AAV expressing an enhanced green fluorescent protein driven by a calcium/calmodulin-dependent kinase II promoter (which is preferentially expressed in glutamatergic neurons) to distinguish between excitatory and inhibitory neurons. In contrast to the above study, there was no difference in mEPSC frequency or amplitude in KO glutamatergic neurons compared to wild-type neurons. However, there were significant increases in mEPSC frequency and amplitude in KO GABAergic neurons compared to wild-type neurons. There were no genotype-dependent differences in mIPSCs in glutamatergic or GABAergic neurons. When looking at evoked responses, the researchers similarly found increased EPSCs in the KO GABAergic neurons compared to WT neurons, but no genotype-dependent differences in evoked IPSCs in GABAergic or glutamatergic neurons. In summary, the authors found a specific increase in excitatory synaptic transmission onto inhibitory interneurons.

Sah et al. [22] also demonstrated that multiple intrinsic properties of KO interneurons, but not glutamatergic neurons, were altered compared to wild-type, suggesting a neuron-specific role for IQSEC2 in development. The KO mice were shown to have increased numbers of PV-positive cells in the hippocampus. In addition, interneurons from wild-type mice were shown to highly express IQSEC2. This is the first report of IQSEC2 expression in inhibitory cells and further supports the role of IQSEC2 in inhibitory transmission.

Using differentiated iPSC cells developed from a child carrying the A350V mutation, Brant et al. [30] showed that immature IQSEC2 mutant dentate gyrus granule neurons were extremely hyperexcitable, exhibiting increased sodium and potassium currents compared to those of CRISPR-Cas9-corrected isogenic controls. Immature IQSEC2 mutant cultured neurons exhibited a marked reduction in the number of inhibitory neurons, which contributed further to hyperexcitability. As the mutant neurons aged, they became hypoexcitable, exhibiting reduced sodium and potassium currents and a reduction in the rate of synaptic and network activity. Jackson et al. [23] also studied the electrophysiological characteristics of neurons from IQSEC2 KO heterozygous females using a microelectrode array. Their results showed that embryonic day 17.5 cultured cortical neurons exhibit hallmarks of immature synaptic networks when compared with their respective wild-type control littermates. These results are similar to those seen in immature A350V differentiated iPSC cells. It is interesting to note that the more mature A350V neurons are more similar to those in the Mehta et al. [21] study, which looked at neurons from older mice (days 14–19). These results point to the additional variable of cellular differentiation that could be a source of discrepancies seen between studies.

A novel finding of the studies described above is that loss of IQSEC2 is shown to affect inhibitory neuronal cell transmission, albeit in opposite directions. Mehta et al. [21], who did not differentiate between excitatory and inhibitory cells, found a decrease in inhibitory as well as stimulatory postsynaptic activity in their KO model, while Sah et al. [22] found an increase only in stimulatory activity in GABAergic neurons. The reason(s) for these differences are not clear; however, the conditions used in the two papers vary in a number of ways, including the use of different IQSEC2 KO mouse models, different types of neurons tested, and different culture conditions. Further studies will be needed to fully understand this newly discovered role of IQSEC2 in inhibitory neurotransmission.

## 7. Increased ARF6-GTP Levels Correlate with IQSEC2 Disease

Shoubridge et al. [1] showed that three mutations in the Sec7 domain of IQSEC2 and one mutation in the IQ region resulted in a decrease in the levels of Arf6-GTP when assayed in a GGA3 pull-down assay of extracts from HEK-293 cells transiently transfected with the mutated genes and Arf6 (see Table 1). Another mutation in the Sec7 domain, A789V, was also shown to result in decreased Arf6-GTP levels [31]. Conversely, in a paper by Rogers et al. [10] the A350V mutant was found to result in increased Arf6-GTP levels. Two other mutations at the same site, A350D and A350T, were also shown to lead to increased Arf6-GTP [32]. Ongoing studies are directed toward understanding how allosteric regulation of the Sec7 domain by the IQ region can explain the unusual effect of the A350 mutants on Arf6-GTP levels in HEK293T cells.

More recent studies using the IQSEC2 mouse KO model reported the paradoxical finding that Arf6-GTP levels are increased in KO brains, despite the absence of IQSEC2 protein [23]. Elevated levels of Arf6-GTP were also seen in wild-type neurons by RNAi-mediated depletion of postsynaptic IQSEC2 (see Table 1) [33]. These studies were conducted in conjunction with cortical neuron cultures and synaptoneurosomes from Fragile X mental retardation protein 1 (FMR1) KO mice. IQSEC2 and Fragile X chromosome disease both cause intellectual disability, developmental delay, and autism. The FMR1 protein normally binds to multiple RNAs, including IQSEC2, and causes an increase specifically in large transcripts such as IQSEC2. In its absence, the levels of IQSEC2 are greatly reduced [34]. Cultured neurons from FMR1 KO mice also displayed increased Arf6-GTP levels [33].

It can be seen from Table 1 that neuronal cell environments in which IQSEC2 is low or absent result in high levels of ARF6-GTP. This situation occurs naturally in immature neurons and can be induced by knocking down the expression of IQSEC2 in mature neurons [8]. The mechanism for this phenomenon is not clear; however, the activation of other GEFs or the inhibition of certain GAPs is likely to be involved. It is interesting to note that elevated levels of activated Arf6 have been reported to exist in a number of dysfunctional situations such as cancer cells. Pharmacological inhibition of guanine nucleotide exchange on Arf6 using SencinH3 or NAV2729 has been used successfully to reduce levels of activated Arf6-GTP in cancer cells and could be a potential therapy for IQSEC2 disease [35].

A child carrying the A350V mutation was observed to have few to no seizures for several weeks after experiencing a high fever. In an attempt to recreate the effect of fever, we placed a child with the A350V mutation in a 40 °C Jacuzzi bath twice daily for 15 min. We found that heat treatments significantly reduced seizures and partially normalized the baseline EEG [36]. This therapeutic effect was recapitulated in a transgenic mouse model of the A350V mutation. It was previously shown that approximately 45% of A350V male mice die between days 16 and 20 after birth due to lethal seizures. Mice incubated at 37 °C starting on day 15 after birth for 5 continuous days showed reduced mortality of 2% [26]. In addition, 2-month-old mutant male mice were shown to be defective in making ultrasonic vocalizations when exposed to wild-type female mice. Mutant mice that received heat treatment on days 15–19 were tested for vocalization capability at two months of age and found to perform at the same level as wild-type mice that had undergone heat treatment [26].

## 8. Heat Treatment Reduces the Seizure Burden and Improves Other Social and Biochemical Defects Seen in A350V IQSEC2 Disease

As mentioned above, HEK293 cells carrying the A350V mutation show increased Arf6-GTP levels. We found that heat treatment at 40 °C resulted in a significant and rapid 50% reduction in Arf6-GTP with no change in total Arf6. This effect was seen for A350D and A350T mutants as well as wild-type IQSEC2 (abstract). We have shown previously that the increase in Arf6-GTP due to the A350V mutation is associated with a reduction in surface expression of AMPA receptors and abnormal spontaneous synaptic transmission in A350V neurons [10]. We found that heat shock (40 °C for 1 h) of A350V murine hippocampal neurons increased surface AMPAR and spontaneous AMPA-dependent EPSCs compared to that seen in WT hippocampal neurons [32].

The mechanism of action of heat therapy in this model may be the activation of the heat shock response (HSR). Hsp90, a major protein induced by the HSR, is known to influence small GTPases such as Ras which could act to reduce the levels of Arf6-GTP [37]. We were able to reproduce the benefit of heat therapy on seizure protection in the A350V transgenic mouse model using celastrol, a chemical inducer of the heat shock response (HSR) [32]. Celastrol reduced the incidence of lethal seizures from 45% to 19%. By contrast, triptolide, an inhibitor of the HSR, abrogated the protective effect of heat treatment, increasing the incidence of lethal seizures in mice receiving heat treatment from 2% to 42% in mice receiving heat treatment and triptolide [32]. These studies point to modulators of the HSR as potential therapeutic agents. Interestingly, communication with parents of IQSEC2 children via an IQSEC2 Facebook group indicates that the phenomenon of fever benefit is apparent in other children carrying different IQSEC2 mutations. Jacuzzi treatments have also been reported to be beneficial in some of these children (personal communication). These reports indicate that heat therapy and/or the induction of the HSR may be applicable to IQSEC2 disease in general. Further studies in this area are needed to work out the details of this mechanism and whether this therapy may be extended to other IQSEC2 mutations.

## 9. Summary and Future Directions

Recent developments in IQSEC2 research include a new role for IQSEC2 in inhibitory neural transmission, the finding that high Arf6-GTP is correlated with the disease, and that heat therapy can reduce seizures and social deficits in children suffering from a mutation in IQSEC2. These findings allow for the development of new therapies for IQSEC2. As mentioned above, Mehta et al. [21] showed for the first time that injection of an AAV carrying the IQSEC2 gene under the control of the EFS promoter into the brain (mPFC) can restore electrophysiological function as well as social ability in KO IQSEC2 mice. This result is encouraging in light of recent FDA-approved AAV gene therapy protocols [38]. More studies using AAV vectors encoding IQSEC2 are needed to substantiate this mode of treatment. It should be noted that some missense mutations may produce a gain of function proteins, as in the case of A350V, which may need to be knocked down in order for gene therapy to succeed.

The testing of small molecules that might benefit IQSEC2 disease has yielded encouraging results [25]. However, continued testing of compounds such as HSR stimulators, GEF inhibitors, and electrophysiological modulators would greatly benefit from a high throughput platform. Several new ideas for IQSEC2 models have emerged for this purpose. These include an IQEC2 KO zebrafish [39] and/or a Xenopus model [40], in which drugs may be tested by dissolving them in the water used to grow the fish and/or tadpoles and tracking them by videography for seizure activity. IQSEC2 is highly conserved in both species and expressed at high levels in the brain. Microelectrode arrays might also be used for screening drugs which might rescue electrophysiological defects seen in KO neurons. Large libraries of FDA-approved small molecules might be tested in this manner. These new directions provide much-needed hope for the future of individuals suffering from IQSEC2 disease.

## Figures and Tables

**Table 1 ijms-24-04984-t001:** Levels of Arf6-GTP as a function of IQSEC2 status and microenvironment.

IQSEC2 Mutation	Environment	Level of Arf6-GTP	Level of IQSEC	Reference
R863W, Q801P, R758Q, R359C	HEK293 cells	Lower than WT	high	[1]
A789V	HEK293 cells	Lower than WT	high	[31]
A350V	HEK293 cells	Higher than WT	high	[10]
IQSEC2 KO male and KO Het female mice	Cortical tissue	Higher than WT	absent	[23]
none	Mouse cortical neuron cultures DIV 21 and adult neurosynaptosomes from Fmr-1 KO mice (Fragile X model)	Higher than WT	low	[33]
none	Rat cortical neurons DIV 21, infected at DIV2 or DIV15 with IQSEC2 KD RNAi	Higher than non-infected neurons	low	[8,33]
none	Rat primary cortical neuron cultures DIV 7	Higher than primary rat cortical neuron cultures DIV 21	low	[8]

## Data Availability

All data reported in this manuscript may be obtained by contacting the authors.

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
