# Peer review of "Molecular Insights into IQSEC2 Disease"

_ijms, 2023, doi:10.3390/ijms24054984_

Round 1
Reviewer 1 Report
The manuscript entitled "Molecular insights into IQSEC2 disease" by Levi et al. reviews the molecular mechanisms of IQSEC2 summarizing phenotypes of mouse models of IQSEC2 diseases, and proposes unique therapeutic protocols.
This review article is helpful to update our knowledge of IQSEC2 diseases, and also give us future insights into the therapy of neuropsychiatric diseases.
My comments (major and minor) are as follows.
Major comments:
1) In line 94-95; because EFS promoter is not "a very strong promoter", rather weak promoter, please change this sentence. The EFS promoter is a broad-spectrum promoter and induces ubiquitous expression of target genes.
2) In line 198; Fragile X mental ratardationprotein 1 for FMR1 is better.
Minor comments:
1) In line 107; "after 21, 27, and 133" should be "after 27, 21, and 133".
2) In the Figure 1, 7th putative domain including R1122C, G1138R, A1139T, and R1155W is better to be shown in the diagram.
3) In line 85; "mcherry-tagged" should be "mCherry-tagged".
4) In line 182; "showed that two mutations" should be "showed that three mutations".
5) In pp. 13, line 25; "sAMPAR" means surface AMPAR, or surface-expressed AMPAR?
Reviewer 2 Report
1. Table 1 is missing.
2. Please add the full name for "KO" abbreviation.
3. Can you add a table summarizing all ~120 mutations reported in IQSEC2 patients.
4. Can you show a figure for western blot analysis showing the mutant A350V protein expressed in the brain.
